# *Ixodes ricinus* Salivary Serpin Iripin-8 Inhibits the Intrinsic Pathway of Coagulation and Complement

**DOI:** 10.3390/ijms22179480

**Published:** 2021-08-31

**Authors:** Jan Kotál, Stéphanie G. I. Polderdijk, Helena Langhansová, Monika Ederová, Larissa A. Martins, Zuzana Beránková, Adéla Chlastáková, Ondřej Hajdušek, Michail Kotsyfakis, James A. Huntington, Jindřich Chmelař

**Affiliations:** 1Department of Medical Biology, Faculty of Science, University of South Bohemia in České Budějovice, Branišovská 1760c, 37005 České Budějovice, Czech Republic; jan.kotal@nih.gov (J.K.); hlanghansova@prf.jcu.cz (H.L.); ederom01@prf.jcu.cz (M.E.); beranz02@prf.jcu.cz (Z.B.); chlasa00@prf.jcu.cz (A.C.); kotsyfakis@paru.cas.cz (M.K.); 2Laboratory of Genomics and Proteomics of Disease Vectors, Institute of Parasitology, Biology Center CAS, Branišovská 1160/31, 37005 České Budějovice, Czech Republic; larissa.martins@nih.gov; 3Cambridge Institute for Medical Research, Department of Haematology, University of Cambridge, The Keith Peters Building, Hills Road, Cambridge CB2 0XY, UK; stephanie.polderdijk@cantab.net (S.G.I.P.); jah52@cam.ac.uk (J.A.H.); 4Laboratory of Vector Immunology, Institute of Parasitology, Biology Center CAS, Branišovská 1160/31, 37005 České Budějovice, Czech Republic; hajdus@paru.cas.cz

**Keywords:** blood coagulation, crystal structure, *Ixodes ricinus*, parasite, saliva, serpin, tick

## Abstract

Tick saliva is a rich source of antihemostatic, anti-inflammatory, and immunomodulatory molecules that actively help the tick to finish its blood meal. Moreover, these molecules facilitate the transmission of tick-borne pathogens. Here we present the functional and structural characterization of Iripin-8, a salivary serpin from the tick *Ixodes ricinus*, a European vector of tick-borne encephalitis and Lyme disease. Iripin-8 displayed blood-meal-induced mRNA expression that peaked in nymphs and the salivary glands of adult females. Iripin-8 inhibited multiple proteases involved in blood coagulation and blocked the intrinsic and common pathways of the coagulation cascade in vitro. Moreover, Iripin-8 inhibited erythrocyte lysis by complement, and Iripin-8 knockdown by RNA interference in tick nymphs delayed the feeding time. Finally, we resolved the crystal structure of Iripin-8 at 1.89 Å resolution to reveal an unusually long and rigid reactive center loop that is conserved in several tick species. The P1 Arg residue is held in place distant from the serpin body by a conserved poly-Pro element on the P′ side. Several PEG molecules bind to Iripin-8, including one in a deep cavity, perhaps indicating the presence of a small-molecule binding site. This is the first crystal structure of a tick serpin in the native state, and Iripin-8 is a tick serpin with a conserved reactive center loop that possesses antihemostatic activity that may mediate interference with host innate immunity.

## 1. Introduction

Ticks are blood-feeding ectoparasites and vectors of human pathogens, including agents of Lyme disease and tick-borne encephalitis. *Ixodes ricinus* is a species of European tick in the Ixodidae (hard tick) family found also in northern Africa and the Middle East [1]. *I. ricinus* ticks feed only once in each of their three developmental stages (larva, nymph, imago), and their feeding course can last over a week in adult females [2]. In order to stay attached to the host for such extended periods of time, ticks counteract host defense mechanisms that would otherwise lead to tick rejection or death.

Insertion of tick mouthparts into host skin causes mechanical injury that immediately triggers the hemostatic mechanisms of blood coagulation, vasoconstriction, and platelet aggregation to prevent blood loss [3]. Consequently, innate immunity is activated as noted by inflammation with edema formation, inflammatory cell infiltration, and itching at tick feeding sites. Long-term feeding and/or repeated exposures of the host to ticks also activate adaptive immunity [4]. As an adaptation to host defenses, ticks modulate and suppress host immune responses and hemostasis by secreting a complex cocktail of pharmacoactive substances via their saliva into the host. For further information on this topic, we refer readers to several excellent reviews describing the impact of saliva and salivary components on the host [4,5,6,7,8].

Blood coagulation is a cascade driven by serine proteases that leads to the production of a fibrin clot. It can be initiated via the extrinsic or intrinsic pathway [9]. The extrinsic pathway starts with blood vessel injury and complex formation between activated factor VII (fVIIa) and tissue factor (TF). The TF/fVIIa complex then activates factor X (fX) either directly or via activation of factor IX (fIX), which in turn activates fX. The intrinsic pathway is triggered by the activation of factor XII (fXII) via kallikrein. Activated fXII (fXIIa) activates factor XI (fXI), which next activates fIX and results in the activation of fX, followed by a common pathway that terminates the coagulation process through the activation of thrombin (fII) and the cleavage of fibrinogen to fibrin, the primary component of the clot [9,10].

Similar to blood coagulation, the complement cascade is based on serine proteases. Complement represents a fast and robust defense mechanism against bacterial pathogens, which are lysed or opsonized by complement to facilitate their killing by other immune mechanisms [11,12]. Complement can be activated via three pathways: the classical pathway, responding to antigen–antibody complexes; the lectin pathway, which needs a lectin to bind to specific carbohydrates on the pathogen surface; and the alternative pathway, which is triggered by direct binding of C3b protein to a microbial surface [12]. All three pathways result in the cleavage of C3 by C3 convertases to C3a and C3b fragments. C3b then triggers a positive feedback loop to amplify the complement response and opsonize pathogens for phagocytosis. Together with other complement components, C3b forms C5 convertase, which cleaves C5 to C5a and C5b fragments. C5b initiates membrane attack complex (MAC) formation, leading to lysis of a target cell. Small C3a and C5a subunits promote inflammation by recruiting immune cells to the site of injury [11].

Both processes, coagulation and complement, are detrimental to feeding ticks, so their saliva contains many anticoagulant and anticomplement molecules, often belonging to the group of serine protease inhibitors (serpins) [13,14,15,16]. Serpins form the largest and most ubiquitous family of protease inhibitors in nature and can be found in viruses, prokaryotes, and eukaryotes [17,18]. Serpins are irreversible inhibitors with a unique inhibitory mechanism and highly conserved tertiary structure [19,20] classified in the I4 family of the MEROPS database [21]. Similar to other serine protease inhibitors, the serpin structure contains a reactive center loop (RCL) that serves as bait for the protease. The RCL amino acid sequence determines serpin’s inhibitory specificity [22].

Arthropod serpins have mostly homeostatic and immunological functions. They regulate hemolymph coagulation or activation of the phenoloxidase system in insects [23]. Additionally, serpins from blood-feeding arthropods can modulate host immunity and host hemostasis [23]. Indeed, over 20 tick salivary serpins have been functionally characterized with described effects on coagulation or immunity [13]. However, according to numerous transcriptomic studies, the total number of tick serpins is significantly higher [13,24,25,26,27]. In *I. ricinus*, at least 36 serpins have been identified based on transcriptomic data, but only 3 of them have been characterized at the biochemical, immunomodulatory, anticoagulatory, or antitick vaccine levels [13,28,29,30,31,32].

Interestingly, one serpin has a fully conserved RCL across various tick species [24]. Homologs of this serpin have been described in *Amblyomma americanum* as AAS19 [33], *Rhipicephalus haemaphysaloides* as RHS8 [34], *Rhipicephalus microplus* as RmS-15 [35], and *I. ricinus* as IRS-8 [30], and it can also be found among transcripts of other tick species in which the serpins have not yet been functionally characterized.

Here we present the functional characterization of Iripin-8, the serpin from *I.*
*ricinus* previously referred to as IRS-8 [30,34], whose RCL is conserved among several tick species. We demonstrate its inhibitory activity against serine proteases involved in coagulation and direct the inhibition of the intrinsic coagulation pathway in vitro. Moreover, we report for the first time the inhibition of complement by a tick serpin. Finally, we provide the structure of Iripin-8 in its native, uncleaved form, revealing an unusual RCL conserved among several tick serpins.

## 2. Results

### 2.1. Iripin-8 Is Predominantly a Salivary Protein with Increased Expression during Tick Feeding

Analysis of Iripin-8 mRNA expression levels revealed its highest abundance in tick nymphs with a peak during the first day of feeding (Figure 1A). In salivary glands, increased Iripin-8 transcription positively correlated with the length of tick feeding on its host. A similar increasing trend was also observed in tick midguts; however, the total number of Iripin-8 transcripts was lower than in the salivary glands. Iripin-8 transcript levels were lowest in the ovaries of all the tested tissues/stages.

Next, we performed Western blot analysis and confirmed the presence of Iripin-8 protein in tick saliva (Figure 1B). We detected two bands of the recombinant protein, representing the full-length native serpin (N) and a molecule cleaved in its RCL near the C-terminus, likely due to bacterial protease contamination (C). The proteolytic cleavage of RCL has previously been documented for serpins from various organisms, including ticks [36,37,38]. The ~5 kDa difference in molecular weight observed between native and recombinant Iripin-8 was probably due to glycosylation, since two N-glycosylation sites are predicted to exist in this serpin. The signal at ~90 kDa in saliva was also detected when using serum from a naïve rabbit (data not shown) and is probably caused by nonspecific antibody binding.

Based on these results, we proceeded to test how Iripin-8 affects host defense mechanisms as a component of tick saliva. Despite the highest expression being observed in the salivary glands, activity in other tissues cannot be ruled out.

### 2.2. Sequence Analysis and Production of Recombinant Iripin-8

The full transcript encoding Iripin-8 was obtained using cDNA from tick salivary glands. Following sequencing, we found a few amino acid mutations (K10 → E10, L36 → F36, P290 → T290, and F318 → S318) compared with the sequence of Iripin-8 (IRS-8) published as a supplement in our previous work [30] (GenBank No. DQ915845.1; ABI94058.1), probably as a result of intertick variability. The RCL was identical to other homologous tick serpins [34], with arginine at the P1 position (Appendix A); however, the remainder of the sequence had undergone evolution, separating species-specific sequences in strongly supported groups (Appendix A). Iripin-8 has a predicted MW of 43 kDa and a pI of 5.85, with two predicted N-linked glycosylation sites.

Iripin-8 was expressed in 2 L of medium with a yield of 45 mg of protein at >90% purity, as analyzed by pixel density analysis in ImageJ software, where a majority was formed from the native serpin and a fraction from a serpin cleaved at its RCL (Appendix A). This mixed sample of native and cleaved serpin was used for all subsequent analyses because the molecules were inseparable by common chromatographic techniques. Proper folding of Iripin-8 was verified by CD spectroscopy (Appendix A) [39,40] and subsequently by activity assays against serine proteases, as presented below. Recombinant Iripin-8 protein solution was tested for the presence of LPS, which was detected at 0.038 endotoxin unit/mL, below the threshold for a pyrogenic effect [41,42].

### 2.3. Iripin-8 Inhibits Serine Proteases Involved in Coagulation

Based on sequence analysis of Iripin-8 and the presence of arginine in the RCL P1 position, we focused on analyzing its inhibitory specificity towards serine proteases related to blood coagulation. Considering the covalent nature of the serpin mechanism of inhibition, we analyzed by SDS-PAGE whether Iripin-8 forms covalent complexes with selected proteases. Figure 2 shows covalent inhibitory complex formation between Iripin-8 and 10 out of 11 tested proteases: thrombin, fVIIa, fIXa, fXa, fXIa, fXIIa, plasmin, APC, kallikrein, and trypsin. We did not detect complexes between Iripin-8 and chymotrypsin. All inhibited proteases could also partially cleave Iripin-8 as indicated by a C-terminal fragment and a stronger signal of cleaved serpin molecule. Chymotrypsin cleaved Iripin-8 in its RCL completely. Inhibition rates of Iripin-8 against these proteases were subsequently determined and are shown in Table 1. Among the tested proteases, plasmin was inhibited significantly faster than other proteases, with a second-order rate constant (*k_2_*) of >200,000 M^−1^ s^−1^. Trypsin, kallikrein, fXIa, and thrombin were inhibited with a *k_2_* in the tens of thousands range and the other proteases with lower *k_2_* values.

### 2.4. Iripin-8 Inhibits the Intrinsic and Common Pathways of Blood Coagulation

Given the in vitro inhibition of coagulation proteases by Iripin-8, we tested its activity in three coagulation assays. The prothrombin time (PT) assay simulates the extrinsic pathway of coagulation, the activated partial thromboplastin time (aPTT) represents the intrinsic (contact) pathway, and thrombin time (TT) represents the final common stage of coagulation. Iripin-8 had no significant effect on PT, which increased from 15.3 to 16.7 s in the presence of 6 μM serpin (not shown). Iripin-8 extended aPTT in a dose-dependent manner, with a statistically significant increase already apparent at 375 nM. With 6 μM Iripin-8, the aPTT was delayed over five times from 31.8 ± 0.4 s to 167.9 ± 3.2 s (Figure 3A). Iripin-8 also inhibited TT in a dose-dependent manner and blocked fibrin clot formation completely at concentrations of 800 nM and higher (Figure 3B). The other serpins presented for comparison in Figure 3C did not have any effect on blood coagulation except the inhibition of PT by Iripin-3, which we published elsewhere [32].

### 2.5. Anticomplement Activity of Iripin-8

The complement pathway readily lyses erythrocytes from various mammals, and those from rabbits were found to be the best complement activators [43]. We used human serum and rabbit erythrocytes to test the effect of tick protease inhibitors on the activity of human complement in vitro. Since the complement cascade is driven by serine proteases, we tested the potential effect of Iripin-8 as a complement regulator. There was a statistically significant reduction in complement activity against erythrocytes when human plasma was incubated with Iripin-8 at concentrations of 2.5 μM and higher (Figure 3C). We also compared Iripin-8 with other two tick salivary serpins: Iripin-3 [32] showed very weak anticomplement activity and was used as a control; compared with Iripin-5 [44], Iripin-8 had lower activity.

### 2.6. Iripin-8 Knockdown Influences Tick Feeding but Not Borrelia Transmission

Since Iripin-8 is predominantly expressed in tick nymphs (Figure 1A), we decided to investigate its importance in tick feeding by RNA interference (RNAi) in the nymphal stage. Knockdown efficiency was 87% for transcript downregulation. Ticks with downregulated Iripin-8 expression showed a significantly lower feeding success rate and higher mortality, with only 51.0% (25/49) finishing feeding compared with 94.1% (48/51) in the control group. Moreover, in ticks that finished feeding, the feeding time was longer compared with control nymphs (Figure 4A). Despite this promising phenotype, we did not observe any effect of Iripin-8 RNAi on the weight of fully engorged nymphs (Figure 4B) or on *B. afzelii* transmission from infected nymphs to mice in any of the tested mouse tissues (Figure 4C).

### 2.7. Role of Iripin-8 in Modulating Host Immunity

Next, we evaluated a possible role for Iripin-8 in the modulation of the host immune response to tick feeding via two assays (OVA antigen-specific CD4^+^ T cell proliferation model using splenocytes isolated from OT-II mice and neutrophil migration towards the chemoattractant (fMLP), in which we previously observed effects with other *I. ricinus* salivary protease inhibitors (the serpin Iripin-3 [32] and the cystatin Iristatin [45]). However, there was no inhibition by Iripin-8 in either assay (Appendix A).

### 2.8. Structural Features of Iripin-8

The crystallographic asymmetric unit contained a single molecule of native Iripin-8 (details in Appendix A). Electron density was of sufficient quality to model all residues from Ser5 to the C-terminus, including the entire RCL. Iripin-8 has the typical native serpin fold with a Cα RMSD (root-mean-square deviation) of 1.93 Å compared with the archetypal serpin alpha-1-antitrypsin (A1AT, 342 of 352 residues, Figure 5A). The most remarkable feature of Iripin-8 is its long RCL (11 residues longer on the P′ side), which extends away from the body of the serpin, moving the P1 Arg364 17.8 Å further than the P1 residue of A1AT. This extended conformation is not the result of a crystal contact; rather it forms the basis of a crystal contact with the RCL of a symmetry-related molecule (Appendix A). The P′ extension contains a stretch of proline residues that form a type II polyproline helix, conferring rigidity and extending the P1 residue away from the body of Iripin-8 (Figure 5B). We can infer from this that the extended RCL is a feature of native Iripin-8 in solution and that it has functional consequences in determining protease specificity and/or inhibitory promiscuity.

We also observed several molecules of PEG (polyethylene glycol) originating from the crystallization buffer bound to Iripin-8. One of the binding sites was a deep 109 Å^3^ cavity in the core structure between helixes A, B, and C (Appendix A). This observation suggests that Iripin-8 can bind small molecules, which may have functional implications. The coordinates and structure factors are deposited in the Protein Data Bank under accession code XXX (note: will be submitted before publication).

## 3. Discussion

Similar to other characterized tick salivary serpins [13], we found that Iripin-8 can modulate host complement and coagulation cascades to facilitate tick feeding [46].

Structurally, Iripin-8 has an unusually long, exposed, and rigid RCL, with an Arg in its P1 position. This potentially enables it to inhibit a range of proteases, as the RCL can interact independently from the body of the serpin molecule. We characterized Iripin-8 as an in vitro inhibitor of at least 10 serine proteases. The interference with the coagulation cascade through inhibition of kallikrein, thrombin, fVIIa, fIXa, fXa, fXIa, and fXIIa in vivo would be beneficial for tick feeding [3,47].

Iripin-8 also inhibited trypsin and kallikrein. Trypsin has a role in meal digestion and has also been linked to skin inflammation [48,49]. Potentially, trypsin inhibition in the host skin could be another mechanism by which the tick impairs the host immune response. Kallikrein has a role in the development of inflammation and pain. It is an activator of the nociceptive mediator bradykinin in the kinin–kallikrein system [50]. Through its inhibition, a deleterious inflammatory response could be altered to the tick’s advantage.

Iripin-8 showed the greatest inhibition of plasmin, a protease involved in fibrin degradation and clot removal [51]. This was surprising, as clot removal should be beneficial for ticks. On the other hand, it is not fully understood whether fibrin clot formation occurs at a tick feeding site in the presence of tick anticoagulant molecules [52]. Apart from fibrinolysis, plasmin also modulates several immunological processes, interacting with leukocytes, endothelial cells, extracellular matrix components, and immune system factors [51,53,54]. Excessive plasmin generation can even lead to pathophysiological inflammatory processes [54]. Considering the proinflammatory role of plasmin, its inhibition by tick salivary serpin could be more relevant to the tick than unimpaired fibrinolysis. Although we did not see any effect of Iripin-8 in two immune assays, we cannot exclude the possibility that Iripin-8 exerts an immunomodulatory effect.

The anticomplement activity of tick saliva or its protein components has been known for decades and is described in numerous publications [14,15,55,56,57]. Although the active molecules originate from either unique tick protein families [58,59,60,61,62] or lipocalins [16], anticomplement activity has only recently been reported for a tick salivary serpin [44] as the only tick protease inhibitor with such activity. Since complement products might directly damage the tick hypostome or initiate a stronger immune response [11], we propose that the role of Iripin-8 is to attenuate these mechanisms. At the same time, an impaired complement system cannot effectively fight pathogens entering the wound at the same time as tick saliva [14]. In this context, we wanted to test a potential effect of Iripin-8 transcriptional downregulation on *Borrelia* transmission from ticks to the host. Although we saw some effect of RNA interference (RNAi) on tick fitness, it had no effect on the amount of *Borrelia* in host tissues. Such a result can be explained by a redundancy in tick salivary molecules, as ticks secrete a variety of effectors against the same host defense mechanism and knockdown of one molecule can be substituted by the activity of others [63].

The increased tick mortality after Iripin-8 knockdown might be due to a potential role for Iripin-8 within the tick body. As an anticoagulant, Iripin-8 can help to keep ingested blood in the tick midgut in an unclotted state for later intracellular digestion [64,65,66]. A similar principle has previously been suggested for other midgut serpins in various tick species [67]. Other potential functions of Iripin-8 include a role in hemolymph clotting [68,69] or in reproduction and egg development [70,71].

The broad inhibitory specificity combined with a conserved, long, and rigid RCL implies that Iripin-8′s role does not necessarily have to only be the modulation of host defense mechanisms. The function of a protruded RCL can be adapted to fit the active site of an unknown protease of tick origin, independently of the serpin body, thus regulating physiological processes in the tick itself, such as melanization and immune processes, which are also regulated by serpins in arthropods.

Interestingly, several PEG molecules from the crystallization buffer bind to Iripin-8, including one in a deep cavity, perhaps indicating the presence of a small-molecule binding site. Considering that serpins can act as transport proteins independently of their inhibitory properties [72,73], the binding properties of Iripin-8 could have physiological relevance in ticks or their tick–host interactions.

By comparing Iripin-8 with other members of the tick serpin group with an identical RCL, we confirmed that the anticoagulant features have also been reported for AAS19 [33] and RmS-15 [35]. RNAi knockdown of Iripin-8 reduced feeding success, while RNAi of AAS19 decreased the blood intake and morphological deformation of ticks [74], and RNAi of RHS8 had an effect on body weight, feeding time, and vitellogenesis [34]. However, these findings are difficult to correlate due to the use of different tick species and life stages. Although Iripin-8 was detected in tick saliva and therefore most likely plays a role in the regulation of host defense mechanisms, further experiments to define Iripin-8 functions in tick tissues would be of interest. Similar to AAS19 [74], Iripin-8 might also regulate hemolymph clotting in the tick body, which is naturally regulated by serpins [67]. Iripin-8 could also contribute to maintaining ingested blood in the tick midgut in an unclotted state to preserve availability for intracellular digestion [64,66].

Although the concentration of Iripin-8 in tick saliva is not known, we can expect it to be lower than most of the concentrations used in our assays. Tick saliva, as a complex mixture, contains an abundance of bioactive molecules that are redundant in their activities and contribute to the inhibition of host defense mechanisms [63]. Therefore, despite the fact that the concentrations of Iripin-8 used in our experiments do not reflect a physiological situation, they can reflect the overall concentration of functionally redundant salivary proteins.

We conclude that the tick serpin Iripin-8 is secreted into the host as a component of *I. ricinus* saliva. Based on its inhibitory activity, mainly of proteases of the coagulation cascade [47], we suggest that its main role as a salivary protein is in the modulation of host blood coagulation and complement activity, with possible function in regulating the immune response. As such, Iripin-8 alters host defense mechanisms and most likely facilitates tick feeding on hosts.

Nevertheless, a more detailed comparative study of tick serpins with conserved RCLs might shed some light on the role of this particular subgroup in different tick species. The conservation of Iripin-8 among tick species suggests a potential for targeting this serpin as a tick control strategy.

## 4. Materials and Methods

### 4.1. Ticks and Laboratory Animals

All animal experiments were carried out in accordance with the Animal Protection Law of the Czech Republic No. 246/1992 Coll., ethics approval No. MSMT-19085/2015-3, and protocols approved by the responsible committee of the Institute of Parasitology, Biology Center of the Czech Academy of Sciences (IP BC CAS). Male and female adult *I. ricinus* ticks were collected by flagging in a forest near České Budějovice in the Czech Republic and kept in 95% humidity chambers under a 12 h light/dark cycle at laboratory temperature. Tick nymphs were obtained from the tick rearing facility of the IP BC CAS. C3H/HeN mice were purchased from Velaz s.r.o. (Prague, Czech Republic). Mice were housed in individually ventilated cages under a 12 h light/dark cycle and used at 6–12 weeks. Laboratory rabbits were purchased from Velaz and housed individually in cages in the animal facility of the Institute of Parasitology. Guinea pigs were bred and housed in cages in the animal facility of the Institute of Parasitology. All mammals were fed a standard pellet diet and given water ad libitum.

### 4.2. Gene Expression Profiling

*I. ricinus* nymphs were fed on C3H/HeN mice for 1 day, 2 days, and until full engorgement (3–4 days); *I. ricinus* females were fed on guinea pigs for 1, 2, 3, 4, 6, and 8 days. Adult salivary glands, midguts, and ovaries, as well as whole nymph bodies, were dissected under RNase-free conditions, and total RNA was isolated using TRI Reagent solution (MRC, Cincinnati, OH, USA). cDNA was prepared using 1 µg of total RNA from pools of ticks fed on three different guinea pigs using the Transcriptor First Strand cDNA Synthesis kit (Roche, Basel, Switzerland) according to the manufacturer´s instructions. The cDNA was subsequently used for the analysis of *Iripin-8* expression by qPCR in a Rotor-Gene 6000 cycler (Qiagen, Hilden, Germany) using FastStart Universal SYBR^®^ Green Master Mix (Roche). *Iripin-8* expression profiles were calculated using the Livak and Schmittgen mathematical model [75] and normalized to *I. ricinus* elongation factor 1α (ef1; GenBank No. GU074829.1) [76,77]. Primer sequences are shown in Appendix A.

### 4.3. RNA Silencing and Borrelia Transmission

*Borrelia afzelii*-infected *I. ricinus* nymphs were prepared as described previously [78,79]. A fragment of the *Iripin-8* gene was amplified from *I. ricinus* cDNA using primers containing restriction sites for ApaI and XbaI (Appendix A; Iripin-8 RNAi) and cloned into the pll10 vector with two T7 promoters in reverse orientations [80]. Double-stranded RNA (dsRNA) of *Iripin-8* and dsRNA of green fluorescent protein (*gfp*) used for control were synthesized using the MEGAscript T7 transcription kit (Ambion, Austin, TX, USA), as described previously [81]. The dsRNA (32 nl; 3 µg/µL) was injected into the hemocoel of sterile or infected nymphs using a Nanoject II instrument (Drummond Scientific, Broomall, PA). After 3 days of rest in a humid chamber at laboratory temperature, ticks were fed on C3H/HeN mice (15–20 nymphs per mouse) until full engorgement. Two weeks later, mice were sacrificed, and the numbers of *Borrelia* spirochetes in the earlobe, urinary bladder, heart tissue, and ankle joint were estimated by qPCR [82] and normalized to the number of mouse genomes [83] (primer and probe sequences in Appendix A). The level of gene knockdown was checked by qPCR in an independent experiment.

### 4.4. Cloning, Expression, and Purification of Iripin-8

The full cDNA sequence of the gene encoding Iripin-8 was amplified with the primers presented in Appendix A using cDNA prepared from the salivary glands of female *I. ricinus* ticks fed for 3 and 6 days on rabbits as a template. The Iripin-8 gene without a signal peptide was cloned into a linearized Champion™ pET SUMO expression vector (Life Technologies, Carlsbad, CA, USA) using NEBuilder^®^ HiFi DNA Assembly Master Mix (New England Biolabs, Ipswich, MA, USA) and transformed into *Escherichia coli* strain Rosetta 2(DE3)pLysS (Novagen, Merck Life Science, Darmstadt, Germany) for expression. Bacterial cultures were fermented in autoinduction TB medium supplemented with 50 mg/L kanamycin at 25 °C for 24 h.

SUMO-tagged Iripin-8 was purified from clarified cell lysate using a HisTrap FF column (GE Healthcare, Chicago, IL, USA) and eluted with 200 mM imidazole. After the first purification, His and SUMO tags were cleaved using a SUMO protease (1:100 *w/w*) overnight at laboratory temperature. Samples were then reapplied to the HisTrap column to separate tags from the native serpin. This step was followed by ion exchange chromatography using a HiTrap Q HP column (GE Healthcare) and by size exclusion chromatography using a HiLoad 16/60 Superdex 75 column (GE Healthcare) to ensure sufficient protein purity.

### 4.5. SDS-PAGE of Complex Formation

Iripin-8 and proteases were incubated at 1 μM final concentrations in a buffer corresponding to each protease (please see below) for 1 h at laboratory temperature. For the assay with fVIIa, we added 1 µM tissue factor (TF). Covalent complex formation was then analyzed in a reducing SDS-PAGE using 4–12% and 12% NuPAGE gels, followed by silver staining.

### 4.6. Determination of Inhibition Constants

Second-order rate constants of protease inhibition were measured by a discontinuous method under pseudo first-order conditions, using at least a 20-fold molar excess of serpin over protease. Reactions were incubated at laboratory temperature and were stopped at each time point by the addition of the chromogenic/fluorogenic substrate appropriate for the protease used. The slope of the linear part of absorbance/fluorescence increase over time gave the residual protease activity at each time point. The apparent (observed) first-order rate constant *k_obs_* was calculated from the slope of a plot of the natural log of residual protease activity over time. *k_obs_* was measured for 5–6 different serpin concentrations, each of them consisting of 8 different time points and plotted against serpin concentration. The slope of this linear plot gave the second-order rate constant *k*_2_. For each determination, the standard error of the mean is given.

The assay buffer was 20 mM Tris, 150 mM NaCl, 5 mM CaCl_2_, 0.2% BSA, 0.1% PEG 8000, pH 7.4 for thrombin, fXa, and fXIa; 20 mM Tris, 150 mM NaCl, 5 mM CaCl_2_, 0.1% PEG 6000, 0.01% Triton X-100, pH 7.5 for activated protein C (APC), fVIIa, fIXa, fXIIa, plasmin, and chymotrypsin; 20 mM Tris, 150 mM NaCl, 0.02% Triton X-100, pH 8.5 for kallikrein and trypsin.

Substrates were: 400 µM S-2238 (Diapharma, Chester, OH, USA) for thrombin; 400 µM S-2222 for fXa (Diapharma); 400 µM S-2366 (Diapharma) for fXIa; 250 µM Boc-QAR-AMC for fVIIa; 250 µM D-CHA-GR-AMC for fXIIa; 250 µM Boc-VPR-AMC for kallikrein, trypsin, and APC; 250 µM D-VLK-AMC for plasmin; and 250 µM Boc-G(OBzl)GR-AMC for fIXa.

Final concentrations and origin of human proteases were as follows: 2 nM thrombin (Haematologic Technologies, Essex Junction, VT, USA), 20 nM fVIIa (Haematologic Technologies), 20 nM TF (BioLegend), 200 nM fIXa (Haematologic Technologies), 5 nM fXa (Haematologic Technologies), 2 nM fXIa (Haematologic Technologies), 10 nM fXIIa (Molecular Innovations, Novi, MI), 8 nM plasma kallikrein (Sigma-Aldrich, St Louis, MO, USA), 1.25 nM plasmin (Haematologic Technologies), 15 nM APC (Haematologic Technologies), 20 pM trypsin (RnD); 10 nM chymotrypsin (Merck).

### 4.7. Anti-Iripin-8 Serum Production and Western Blotting

Serum with antibodies against Iripin-8 was produced by immunization of a rabbit with pure recombinant protein as described previously [84]. Tick saliva was collected from ticks fed for 6 days on guinea pigs by pilocarpine induction as described previously [85]. Tick saliva was separated by reducing electrophoresis using NuPAGE™ 4–12% Bis-Tris gels. Proteins were either visualized using Coomassie staining or transferred onto PVDF membranes (Thermo Fisher Scientific). Subsequently, membranes were blocked in 5% skimmed milk in Tris-buffered saline (TBS) with 0.1% Tween 20 (TBS T) for 1 h at laboratory temperature. Membranes were then incubated with rabbit anti-Iripin-8 serum diluted in 5% skimmed milk in TBS-T (1:100) overnight at 4 °C. After washing in TBS-T, the membranes were incubated with secondary antibody (goat anti-rabbit) conjugated with horseradish peroxidase (Cell Signaling Technology; Danvers, MA, USA; 1:2000). Proteins were visualized using the enhanced chemiluminescent substrate WesternBright^TM^ Quantum (Advansta, San Jose, CA, USA) and detected using a CCD imaging system (Uvitec, Cambridge, UK).

### 4.8. Coagulation Assays

All assays were performed at 37 °C using preheated reagents (Technoclone, Vienna, Austria). Normal human plasma (Coagulation Control N) was preincubated with Iripin-8 for 10 min prior to coagulation initiation. All assays were analyzed using the Ceveron four coagulometer (Technoclone).

For prothrombin time (PT) estimation, 100 μL plasma was preincubated with 6 μM Iripin-8, followed by the addition of 200 µL Technoplastin^®^ HIS solution and estimation of fibrin clot formation time. For activated partial thromboplastin time (aPTT), 100 μL plasma was preincubated with various concentrations of Iripin-8 (94 nM–6 μM), followed by the addition of 100 μL of Dapttin^®^ TC and incubation for 2 min. Coagulation was triggered by the addition of 100 μL 25 mM CaCl_2_ solution. For thrombin time (TT), 200 μL of thrombin reagent was incubated with various concentrations of Iripin-8 for 10 min and subsequently added to 200 μL of plasma to initiate clot formation.

### 4.9. Crystal Structure Determination

Iripin-8 was concentrated to 6.5 mg/mL and dialyzed into 20 mM Tris pH 7.4, 20 mM NaCl. Crystals were obtained from the PGA screen [86] (Molecular Dimensions, Maumee, OH) in 0.1 M Tris pH 7.8, 5% PGA-LM, 30% *v/v* PEG 550 MME. Crystals were flash-frozen in liquid nitrogen straight from the well condition without additional cryoprotection. Data were collected at the Diamond Light Source (Didcot) on a beamline I04-1 and processed using the CCP4 suite [87] as follows: integration by Mosflm [88] and scaling and merging with Aimless [89]. The structure was solved by molecular replacement with Phaser [90]. The template for molecular replacement was generated from the structure of conserpin (PDB ID 5CDX [91]), which was truncated to remove flexible regions and mutated using Chainsaw [92] based on a sequence alignment to Iripin-8 using Expresso [93]. The structure was refined with Refmac [94]. Model quality was assessed by MolProbity [95,96], and figures were generated using PyMOL [97]. 

### 4.10. Complement Assay

Fresh rabbit erythrocytes were collected in Alsever’s solution from the rabbit marginal ear artery, washed three times in excess PBS buffer, and finally diluted to a 2% suspension (*v/v*). Fresh human serum was obtained from three healthy individuals. The assay was performed in a 96-well round-bottomed microtiter plate (Nunc, Thermo Fisher Scientific). Each well contained 100 μL 50% human serum in PBS premixed with different concentrations of Iripin-8 (315 nM–10 µM). After 10 min incubation at laboratory temperature, 100 μL of erythrocyte suspension was added (i.e., 25% final serum concentration after the addition of erythrocyte suspension to a final 1%). Reaction wells were observed individually under a stereomicroscope using oblique illumination and an aluminum pad, and the time needed for erythrocyte lysis was measured. When full lysis was achieved, the reaction mixture turned from opaque to transparent. Negative controls did not contain either Iripin-8 or human serum. Additional controls were performed with heat-inactivated serum (56 °C, 30 min). The assay was evaluated in technical and biological triplicates.

### 4.11. Immunological Assays

Both the CD4^+^ T cell proliferation assay and neutrophil migration assay were performed following the protocols described by Kotál et al. [45]. Briefly, for the CD4^+^ T cell proliferation assay, splenocytes were isolated from OT-II mice, fluorescently labeled, pre-incubated with serpin for 2 h, and their proliferation stimulated by the addition of OVA peptide. After 72 h, cells were labelled with anti-CD4 antibody and analyzed by flow cytometry. For the migration assay, neutrophils were isolated from mouse bone marrow by immunomagnetic separation and preincubated with serpin for 1 h. Cells were then seeded in the inserts of 5 μm pore Corning^®^ Transwell^®^ chambers (Corning, Corning, NY, USA) and were allowed to migrate towards an fMLP (Sigma-Aldrich) gradient for 1 h. The migration rate was determined by cell counting using the Neubauer chamber.

### 4.12. Statistical Analysis

All experiments were performed as three biological replicates. Data are presented as mean ± standard error of mean (SEM) in all graphs. Student’s *t*-test or one-way ANOVA was used to calculate statistical differences between two or more groups, respectively. For RT-PCR, data for nymphs, salivary glands, midgut, and ovaries were analyzed separately using one-way ANOVA, followed by Dunnett’s post hoc test. Statistically significant results are marked: * *p* ≤ 0.05; ** *p* ≤ 0.01; *** *p* ≤ 0.001; n.s., not significant.

## Figures and Tables

**Figure 1 ijms-22-09480-f001:**
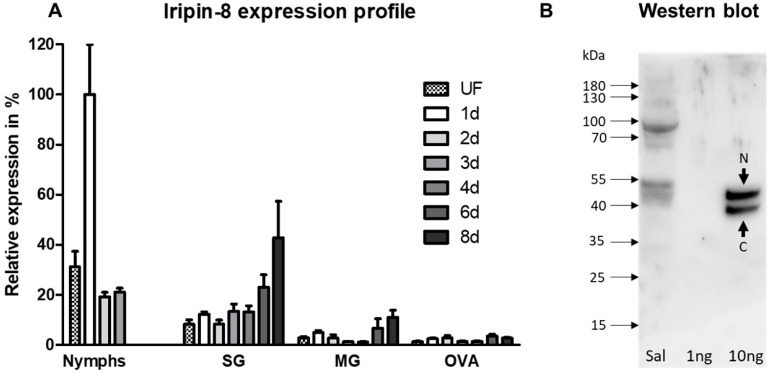
Iripin-8 expression in ticks and its presence in tick saliva. (**A**) Pools of *I. ricinus* salivary glands, midguts, and ovaries from female ticks and whole bodies from nymphs were dissected under RNase-free conditions. cDNA was subsequently prepared as a template for qRT-PCR. Iripin-8 expression was normalized to elongation factor 1α and compared between all values with the highest expression set to 100% (*y*-axis). The data show an average of three biological replicates for adult ticks and six replicates for nymphs (±SEM). SG = salivary glands; MG = midguts; OVA = ovaries; UF = unfed ticks; 1 d, 2 d, 3 d, 4 d, 6 d, 8 d = ticks after 1, 2, 3, 4, 6, or 8 days of feeding. For nymphs, the last column represents fully fed nymphs. All feeding points for each development stage/tissue are compared with the unfed ticks of the respective group. (**B**) Iripin-8 can be detected in tick saliva by Western blotting. Saliva from ticks after 6 days of feeding and recombinant Iripin-8 protein were visualized by Western blotting using serum from naïve and Iripin-8-immunized rabbits. Sal = tick saliva; 1 ng, 10 ng = Iripin-8 recombinant protein at 1 ng and 10 ng load. N: native Iripin-8, C: cleaved Iripin-8.

**Figure 2 ijms-22-09480-f002:**
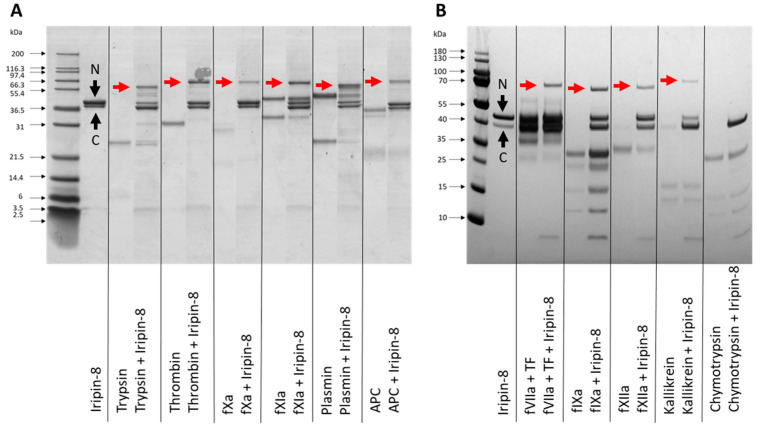
Formation of covalent complexes between Iripin-8 and serine proteases. Iripin-8 and selected serine proteases were incubated for 1 h and subsequently analyzed for complex formation by reducing SDS-PAGE. Protein separation differs between (**A**,**B**) due to the use of gels with different polyacrylamide contents. Gels show the profile of Iripin-8 serpin alone, various serine proteases alone, and proteases incubated with Iripin-8. Complex formation between fVIIa and Iripin-8 was tested in the presence of tissue factor (TF) at an equimolar concentration. Covalent complexes between Iripin-8 and protease are marked with a red arrow. N: native Iripin-8, C: cleaved Iripin-8.

**Figure 3 ijms-22-09480-f003:**
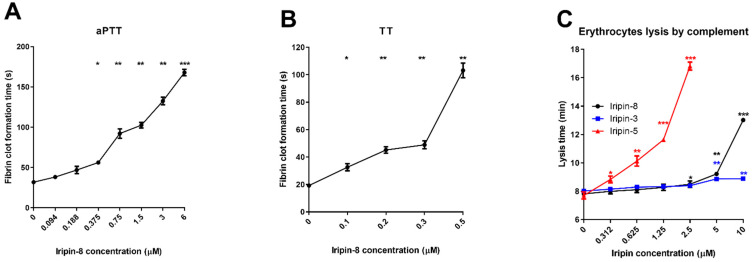
Inhibition of complement and coagulation pathways by Iripin-8. (**A**) Iripin-8 inhibits the intrinsic coagulation pathway. Human plasma was preincubated with increasing concentrations of Iripin-8 (94 nM–6 μM). Coagulation was triggered by the addition of Dapttin^®^ reagent and CaCl_2_, and clot formation time was measured. A sample without Iripin-8 was used as a control for statistical purposes. (**B**) Iripin-8 delays fibrin clot formation in a thrombin time assay in a dose-dependent manner. Coagulation of human plasma was initiated by thrombin reagent preincubated with various concentrations of Iripin-8, and thrombin time was measured. Samples without Iripin-8 were used as a control for statistical purposes. (**C**) Iripin-8 inhibits erythrocyte lysis by human complement. Human plasma was preincubated with increasing concentrations of Iripin-3, 5, and 8 (312 nM–10 μM). After the addition of rabbit erythrocytes, their lysis time by complement was measured. Values represent prolongation of time needed for erythrocyte lysis compared with the control group. * *p* ≤ 0.05; ** *p* ≤ 0.01; *** *p* ≤ 0.001.

**Figure 4 ijms-22-09480-f004:**
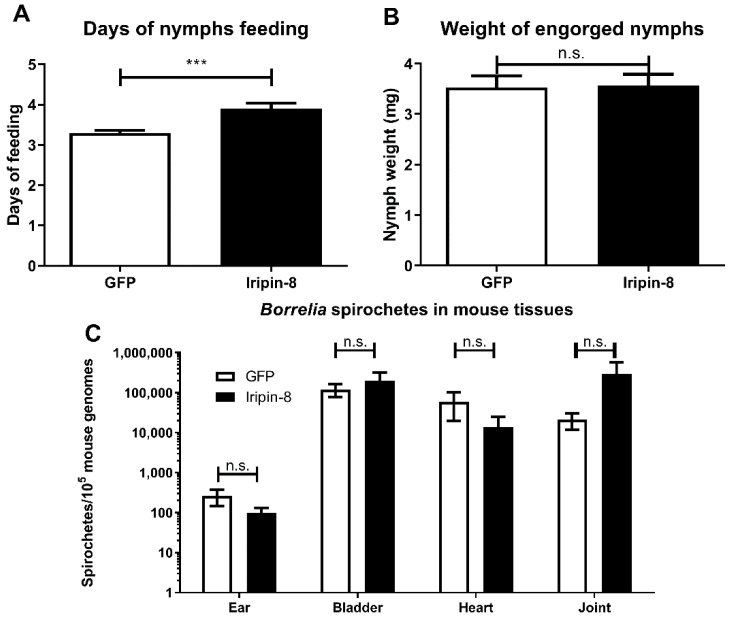
Effect of RNAi on tick fitness and *Borrelia* transmission. (**A**) RNAi of Iripin-8 prolonged the length of *I. ricinus* nymph feeding compared with the control group (GFP). (**B**) Weight of fully engorged nymphs with Iripin-8 knockdown was not different from the control group (GFP). (**C**) Presence of *B. afzelii* spirochetes in mouse tissues after infestation with infected *I. ricinus* nymphs. There were no significant differences between Iripin-8 knockdown and GFP control groups in any of the tested tissues. *** *p* ≤ 0.001; n.s., not significant.

**Figure 5 ijms-22-09480-f005:**
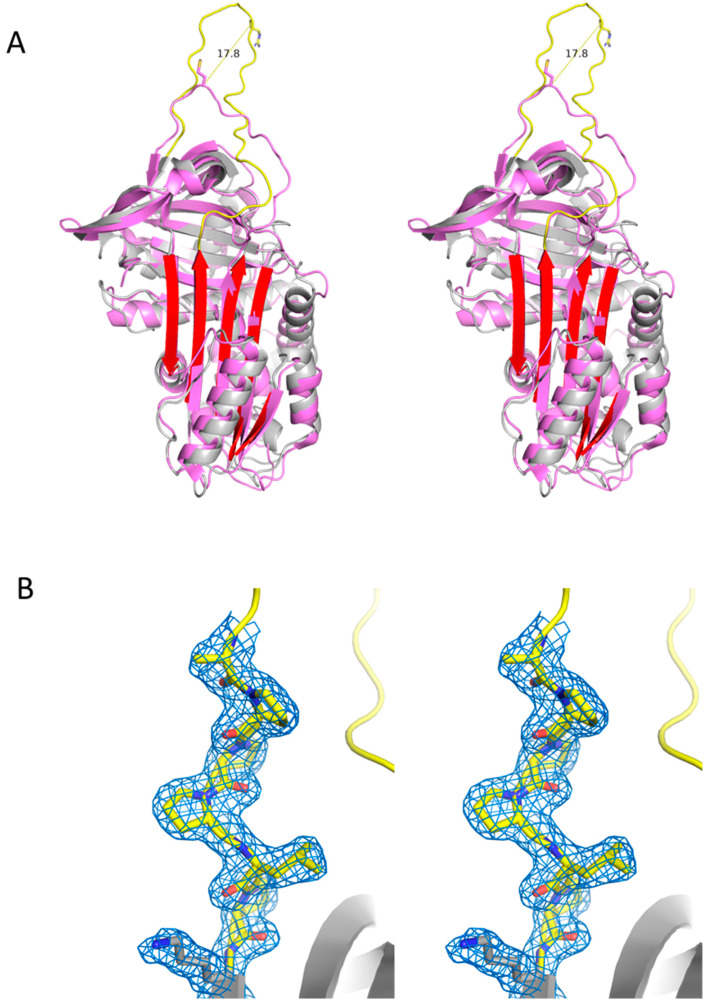
Crystal structure of native Iripin-8. (**A**) Stereo view of a ribbon diagram of Iripin-8 (gray with yellow RCL and red beta sheet A) superimposed with alpha-1-antitrypsin (PDB code 3ne4). The P1 side chains of both molecules are represented as sticks, and the distance between their Cα atoms is shown. (**B**) Stereo view of a close-up of the P′ region with surrounding electron density (contoured at 1 times the RMSD of the map), forming a rigid type II polyproline helix.

**Table 1 ijms-22-09480-t001:** Inhibition rate of Iripin-8 against selected serine proteases.

Protease	*k*_2_ (M^−1^ s^−1^)	±SE
Plasmin	225,064	14,183
Trypsin	29,447	3508
Kallikrein	16,682	1119
fXIa	16,328	948
Thrombin	13,794	1040
fXIIa	3324	409
fXa	2088	115
APC	523	35
fVIIa + TF	456	35
fIXa	N/A	N/A

## Data Availability

All data are either contained within the manuscript and supporting information or available from the corresponding author on reasonable request.

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
