# Peer review of "Ixodes ricinus* Salivary Serpin Iripin-8 Inhibits the Intrinsic Pathway of Coagulation and Complement"

_ijms, 2021, doi:10.3390/ijms22179480_

Round 1

Reviewer 1 Report

Review manuscript International Journal of Molecular Sciences

Titel: Ixodes ricinus salivary serpin Iripin-8 inhibits the intrinsic pathway of coagulation and complement

Authors: Jan Kotál, et al.                                               

In this manuscript, Kotál and co-workers, unraveled the functional and structural characteristics of Iripin-8, a salivary serpin from the tick Ixodes ricinus.

Iripin-8 displayed blood meal-induced mRNA expression that peaked in nymphs and the salivary glands of adult females. It appeared that Iripin-8 inhibited multiple proteases involved in blood coagulation and blocked the intrinsic and common pathways of the coagulation cascade in vitro. In addition, Iripin-8 inhibited erythrocyte lysis by complement, and Iripin-8 knockdown by RNA interference in tick nymphs delayed the feeding time. Structural elucidation with X-ray crystallography showed that  Iripin-8 is a tick serpin with a conserved, but unusual long and rigid reactive center loop.

According to the reviewer, the objectives of this study are clear, the setup of the experiments in the manuscript are in general sufficient and the results generally support the conclusions.

Comments:

For this reviewer it is not clear how Iripin-8 is cleaved, resulting in native Iripin-8 (N) and cleaved Iripin-8 (C). Is this due to enzymatic degradation? Or is this an alternative expressed form? Or is this a product of protein instability? Please, make this clearer in the manuscript.

Are in Figure 2 and Table 1 the results presented for native Iripin-8 (N) or for cleaved Iripin (C)? This is not clear.

Do the authors probably have an explanation for the fact that Iripin-8 is highly expressed after 2 days of feeding, whereas it takes longer (8 days) to be highly expressed in the salivary glands of adults?

The authors present clearly the erythrocyte lysis time for Iripin-3, -5 and -8 in Figure 3C. It would be good to present the results of Iripin-3 and -5 in Figure 3A and C as well. Then, it is clear at a glance what these proteins do in the aPTT, TT, (and PT) assays as well.

An alignment of AAS19, RHS8, RmS-15, and Iripin-8 (IRS-8) would be helpful (if there is room in a Figure).

Now, it is not clear which complement pathway is inhibited by Iripin-8. Is there a possibility to unravel this?

Reviewer 2 Report

Authors of this manuscript performed characterization of Iripin 8, one of the inhibitors of serine proteases (serpins) present in the tick I. ricinus saliva. They confirmed presence of Iripin 8 in saliva, prepared recombinant Iripin 8, and performed its functional characterization and structural studies. Manuscript contains novel data, e.g. the crystal structure and biochemical characterization, formation of covalent complexes of Iripin 8 with proteases and their inhibition. They tested a potential anti-complement function of Iripin 8. It the present manuscript there are however, several technical issues, which need to be added. Most notably, there is a lack of any statistics and error bars in Figure 3. Table 1 with inhibition rates must contain standard deviations and legend should describe how many kinetics experiments were measured to get these values.

Fig. 1: Two bands were detected on WB illustrating the presence of Iripin 8 in tick saliva as well as in purified recombinant Iripin 8. These bands are marked as N- native, and C- cleaved Iripin 8. It is not clear, what does represent the cleaved band. It is necessary to characterize the band „C“ either by N-terminal sequencing or MS analysis, or previously published citation (if there is any).

Authors describe unusual length of reactive center loop (RCL) and present alignment in supplementary Figure 1. It is difficult to read in this figure size even in the electronic form. I recommend to present SFig.1A in the size of full column and Phylogenetic tree place under this alignment. Moreover, mark with a frame or color Iripin 8 in the tree.

Discussion should contain information about biological concentration of iripins in tick saliva and relevance of inhibition concentrations of Iripin 8 from in vitro experiments with a potential effect in vivo.
